# Mitochondrial genome refinement and comparative phylogenetics of *Parastrongyloides trichosuri* (KNP strain; Nematoda: Strongyloididae) from South Africa

**Kwangjae Cho[1], Minkyung Kim[1], Jun Won Park[1], Yang-Kyu Choi[2], Won Gi Yoo[1]/+**

[1]Seoul National University, College of Veterinary Medicine and Research Institute for Veterinary Science, Seoul, Republic of Korea
[2]Konkuk University, College of Veterinary Medicine, Department of Laboratory Animal Medicine, Seoul, Republic of Korea

**BACKGROUND** *Parastrongyloides trichosuri* Mackerras, 1959 (Nematoda: Strongyloididae) is a facultatively parasitic nematode infecting the common brushtail possum. A previously reported mitochondrial genome (mitogenome) from a Kruger National Park (KNP) isolate (GenBank: NC_028620) is incomplete, lacking the *nad3* gene and the noncoding region (NCR), limiting its utility for comparative and phylogenetic studies.

**OBJECTIVES** This study aimed to reconstruct, annotate, and validate a complete mitogenome of *P. trichosuri* (KNP strain) to enhance genomic accuracy and phylogenetic resolution within Strongyloididae.

**METHODS** Whole-genome sequencing data (SRA: ERS056619) were reanalysed. Missing genes, including tRNA-Glu and *atp6*, were manually curated using BLASTn searches against Rfam v15, while *cox3* and *nad3* were confirmed through transmembrane topology analysis.

**FINDINGS** The reconstructed mitogenome was 13,809 bp long, comprising 12 protein-coding genes, 22 tRNAs, 2 rRNAs, and a 539-bp tandem-repeat NCR. Gene order and structure were consistent with other Strongyloididae mitogenomes. Phylogenetic analysis supported *P. trichosuri* as a distinct lineage within the family. The annotated sequence has been deposited in the Third Party Annotation database in GenBank (accession No. BK075097).

**MAIN CONCLUSIONS** This improved mitogenome fills an existing genomic gaps and provides a reliable reference for future comparative, phylogenetic, and evolutionary studies of Strongyloididae.

Key words: *Parastrongyloides trichosuri* - mitogenome - Strongyloididae - reannotation - reassembly

The genus *Parastrongyloides* Morgan, 1928 (Nematoda: Strongyloididae) comprises facultatively parasitic nematodes that exhibit both free-living and parasitic generations within their life cycle.[1] Among them, *Parastrongyloides trichosuri* Mackerras,1959 first described from the brush-tailed possum (*Trichosurus vulpecula*) in Australia, has become a key experimental model for understanding the transition between free-living and parasitic life styles in nematodes. The species is phylogenetically related to *Strongyloides* spp., sharing the ability to produce free-living adults and infective third-stage larvae that develop in the environment and resume parasitic growth upon infection of a suitable mammalian host.[2]

Obtaining and annotating the mitochondrial genome (mitogenome) of *P. trichosuri* is essential for several reasons. First, it will provide baseline genomic information to compare gene content, order, and nucleotide composition with other Rhabditid and Strongyloid nematodes, shedding light on the evolutionary origin of facultative parasitism.[2] Second, mitogenomic phylogenies may refine the placement of *Parastrongyloides* relative to *Strongyloides* and allied taxa, complementing nuclear-gene analyses that remain unresolved.[3] Finally, a well-annotated mitogenome will enrich comparative datasets for elucidating gene-order evolution within Chromadorea and support population-genetic and evolutionary-developmental studies.[4]

Collectively, these perspectives highlight the significance of reconstructing a complete mitogenome of *P. trichosuri*, which will address a major gap in nematode comparative genomics and enhance our understanding of the evolution of parasitism within the Strongyloididae. Although a mitogenome of *P. trichosuri* from Kruger National Park (KNP), South Africa, has previously been deposited in GenBank (accession No. NC_028620; Pt-i), it represents an incomplete sequence lacking both the *nad3* gene and the noncoding region (NCR). In this study, we report a complete mitogenome (Pt-c) of *P. trichosuri* that has been reconstructed and accurately annotated.

**doi:** 10.1590/0074-02760250294

**Financial support:** This work was supported by the KNRF (grants: RS-2022-NR070067 and RS-2024-00509361), RIVS, SNU, Korea.
**+ Corresponding author:** wongi.yoo@snu.ac.kr | ⓘ https://orcid.org/0000-0002-3041-5560

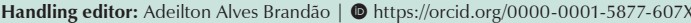

**Handling editor:** Adeilton Alves Brandão | ⓘ https://orcid.org/0000-0001-5877-607X

## MATERIALS AND METHODS

*Retrieval and preprocessing of genomic data from the NCBI database* - Whole-genome sequence (WGS) data of *P. trichosuri* were retrieved from the '50 Helminth Genomes Project'.[2] The Illumina paired-end reads (2 × 100 bp) were downloaded in the Sequence Read Archive (SRA) under the accession No. ERS056619. Briefly, the raw reads (41.43 Gb each paired-end read sample) were generated using Illumina HiSeq 2000. A total of 38.14 Gb of clean data per paired-end read sample was obtained after adapter removal and quality trimming (Phred score cutoff of 33) using Trimmomatic v0.39.[5]

*Reads filtering and de novo assembly* - The cleaned data were used to reconstruct the complete mitogenome assembly of *P. trichosuri*. *De novo* assembly was performed with GetOrganelle toolkit v1.7.7.1[6] which has been reported as one of the best-performing assembly pipelines for mitogenome reconstruction.[7] The seed/label sequence databases were customised using Nematoda RefSeq mitogenomes (Taxonomy ID: 6231). Pre-filtering was applied using the Nematoda RefSeq sequences to enrich mitochondrial reads. The resulting filtered reads were taxonomically classified with Kraken 2 (July 2025 release)[8] to evaluate potential host or pathogen contamination. The final assembly was generated using the built-in SPAdes v4.2.0[9] assembler, and quality validation was conducted by remapping the filtered reads with Bowtie 2 v2.5.4[10] to ensure the absence of detectable contamination. The assembly graph was further inspected and adjusted using Bandage v0.8.1.[11]

*Functional genome annotation and identification of missing genes* - Mitogenome annotation, including the mitochondrial protein-coding genes (mPCG) and functional RNAs, such as transfer RNA (tRNA) and ribosomal RNA (rRNA), was performed using the MITOS2 pipeline[12] through the GALAXY server v2.1.9 (https://usegalaxy.org/). The open reading frames (ORFs) of the mPCGs were manually confirmed using the ORF finder (https://www.ncbi.nlm.nih.gov/orffinder/) with the invertebrate mitochondrial genetic code and were identified by comparison with the reported mitogenomes of the family Strongyloididae. tRNA-Glu (*trnE*) and *apt6* not detected by MITOS2 were recovered through BLASTn searches against Rfam v15 containing Strongyloididae sequences.[13] For *cox3* and *nad3*, both ORF and distinctive transmembrane (TM) domains were identified using TMHMM v2.0[14] and DeepTMHMM v1.0 (https://services.healthtech.dtu.dk/services/DeepTMHMM-1.0/). Repeat units (RUs) were predicted using Tandem Repeats Finder v4.09.1[15] with default parameters. All gene organisation features were visualised using Proksee v6.0.2.[16] The Dynamic Genomic Alignment server (DiGAlign) v2.0[17] was employed with default parameters to compare synteny and perform alignment of genomic elements via BLASTn.

*Mitochondrial phylogenomics* - A total of 12 mPCGs, as well as all individual genes, were retrieved for nine Strongyloididae taxa (including *P. trichosuri*) from NCBI GenBank [Supplementary data (Table)].

Amino acid sequences were obtained in FASTA format and concatenated for the mPCG dataset. Each dataset including concatenation of 12 mPCGs (con-mPCGs) and all individual genes was aligned at the amino acid level using MAFFT v7.505[18] with default parameters. Ambiguously aligned or poorly conserved regions were manually inspected and removed in MEGA11[19] prior to phylogenetic analysis. Phylogenetic trees were inferred under the maximum likelihood framework in MEGA11. For each dataset, the Le and Gascuel substitution model with a discrete Gamma distribution to account for among-site rate variation (five categories) was applied. The shape parameter (α) was estimated from the data. Initial trees for the heuristic search were generated by applying the Neighbor-Joining and BioNJ algorithms to a matrix of pairwise distances estimated using the JTT model, and the topology with the highest log likelihood was selected. Branch support was assessed by bootstrap analysis with 1,000 pseudoreplicates. Two *Schistosoma* species were used as outgroups, specifically *Schistosoma mansoni* and *S. japonicum*, as suggested by Viney [Supplementary data (Table)].[20]

## RESULTS

*Completion of P. trichosuri mitogenome assembly and annotation* - Reconstruction of *P. trichosuri* mitogenome from WGS data was performed using GetOrganelle toolkit,[6] which applies preassembly filtering based on organelle-specific seed sequences. Taxonomic classification of the filtered reads using Kraken 2[8] revealed that no mammalian host DNA contamination was present. However, bacterial contaminants accounted for 48.40% of the total reads, with the majority belonging to the phylum *Pseudomonadota* (38.76%) [Supplementary data (Fig. 1)]. After *de novo* assembly with the built-in SPAdes[9] assembler, and subsequent validation by Bowtie 2[10] remapping, no detectable bacterial sequences remained in the final assembly. The assembly graph, inspected in Bandage,[11] revealed a single circular contig, with no ambiguous or unresolved regions detected [Supplementary data (Fig. 2)].

The *trnE* and *atp6* were manually annotated using BLASTn searches against the Strongyloididae sequences. In particular, *cox3* and *nad3* were successfully refined and validated through comparative analysis of TM domain topology, which confirmed that *cox3* possesses seven TM domains and *nad3* three ones, respectively (Fig. 1A-B). The resulting well-annotated circular mitogenome measured 13,809 bp in length (Table I, Fig. 1).

*Structural and compositional features of the complete mitochondrial genome* - Complete and incomplete mitogenomes of *P. trichosuri* were displayed in circular maps showing 36 genes and 35 genes, respectively [Fig. 1C, Supplementary data (Fig. 3)]. The complete mitogenome (Pt-c) of *P. trichosuri* includes 12 mPCGs (*nad1-6, nad4L, cox1-3, cytb, atp6*), 2 rRNA genes (*rrnL, rrnS*), and 22 tRNA genes, which is consistent with the mitogenomes of the other *Strongyloides* species (Table I).

The total length of the mPCGs was 10,252 bp, which is a proportion of 74.25% of the entire complete mitogenome, compared to 11,567 bp (84.62%) of the incomplete

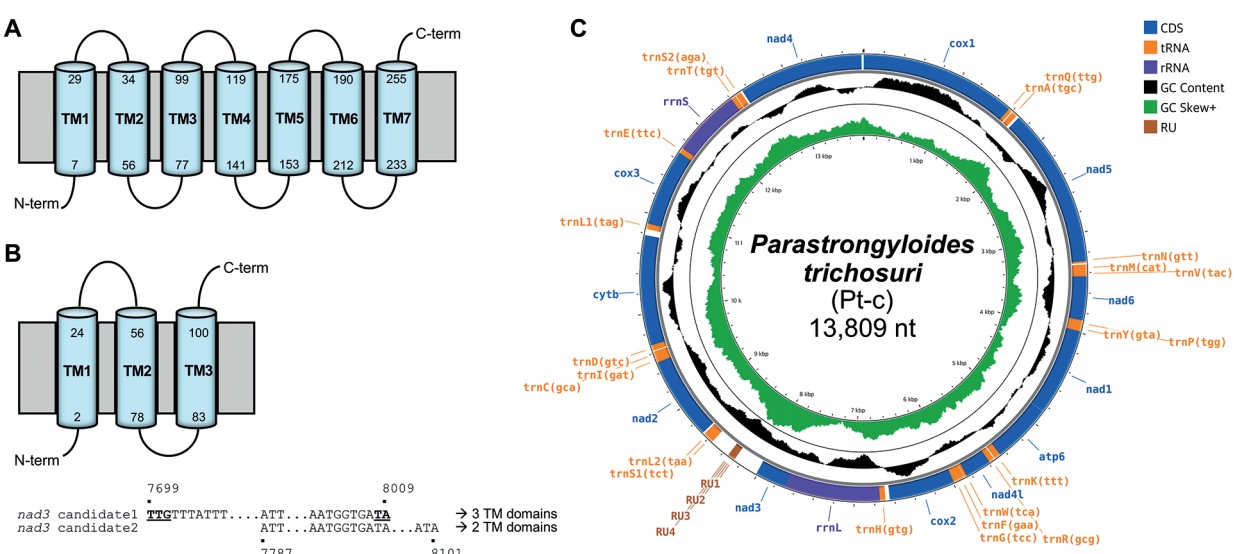

Fig. 1: complete mitogenome (Pt-c, BK075097) of *Parastrongyloides trichosuri* showing annotated genomic features. Predicted transmembrane (TM) topologies of *cox3* (A) and *nad3* (B), respectively, showing complete agreement between TMHMM v2.0 and DeepTMHMM v1.0 in the number and position of predicted TM domains. The final *nad3* sequence was determined based on pairwise alignment of two candidates and the consistency of their predicted TM domain counts. Start and stop codons are underlined. (C) Final circular map of the mitogenome.

mitogenome (Pt-i). *nad5* (1,596 bp) was the longest while *nad4L* (289 bp) was the shortest of mPCGs. TTG and TAA were the most prevalent codons for initiation and termination, respectively. The whole length of tRNA genes was 1,272 bp, with the length ranging from 52 nucleotides (*trnS2*) to 83 nucleotides (*trnL2*) in the complete mitogenome (Pt-c). Compared with the incomplete mitogenome (Pt-i), the complete mitogenome (Pt-c) revealed the precise annotation of *cox3* and the presence of a 539-bp NCR containing a 75-bp segment composed of three 20-bp RUs and an additional 15-bp short repeat sequence [Table I, Supplementary data (Fig. 4)].

*Comparative mitogenomic analysis of P. trichosuri and other Strongyloides species* - The genetic differences of 12 mPCGs of the mitogenomes of representative species in the family Strongyloididae were compared at the nucleotide level (Table II). The sequence differences between the complete mitogenome (Pt-c) of *P. trichosuri* and the other eight *Strongyloides* species ranged from 27.31% (*S. ratti*) to 43.87% (*S. papillosus*). Interestingly, *S. papillosus* exhibited the greatest divergence across all individual genes. Large differences in sequences were detected in *nad6* (63.32%), *atp6* (63.23%), *nad4L* (57.75%), *nad3* (53.94%), and *nad2* (53.12%).

Mitogenome-based phylogenetic analysis of *P. trichosuri* and eight *Strongyloides* species revealed two distinct clades, in which *P. trichosuri* formed clade I together with *S. cebus*, *S. stercoralis* and *S. ratti* (Fig. 2). All mitogenomes were compared by automatic repositioning based on gene synteny similarities. Among them, the *P. trichosuri* mitogenome showed extensive rearrangements, consistent with the patterns observed in other *Strongyloides* species. Within clade I, the *cox1* gene was highly conserved, whereas *cytb*, *nad1*, *nad4*, *nad5*, two rRNA genes, and *trnH* exhibited nucleotide identities ranging from 70% to 85%.

*Phylomitogenic analysis* - Phylogenetic relationships are well resolved with high nodal support (Fig. 3). *P. trichosuri* showed a close relationship with three *Strongyloides* speces, such as *S. ratti*, *S. cebus*, and *S. venezuelensis*. Phylogenetic topologies inferred from *apt6*, *nad2-4*, *nad4L*, and *nad6* were largely congruent, consistently by revealing two distinct clades. These gene-based phylogenies were consistent with the mitogenome-based phylogenetic tree. Interestingly, the *cox2*-based phylogeny showed *P. trichosuri* forming a distinct group with *S. ratti*, which corresponded to the lowest nucleotide divergence (19.93%) among all mPCGs (Table II).

## DISCUSSION

Mitogenome is informative and robust molecular markers for exploring helminth systematics, population structure, and evolutionary biology.[21] However, helminth mitogenomes, particularly those of cestodes, exhibit substantial variability in assembly completeness including truncated protein-coding regions or unresolved NCR, and annotation accuracy including missing tRNA genes.[22,23] Similarly, several *Strongyloides* mitogenomes deposited in the NCBI Reference Sequence Database (RefSeq) remain incomplete, such as *S. papillosus* (NC_028622) missing *nad4*, *S. venezuelensis* (NC_028229) and *S. ratti* (NC_028623) missing *trnV*, *S. stercoralis* (NC_028624) missing *trnD*, and *S. cebus* (NC_066659) showing a duplicated *trnN* [Supplementary data (Table)], lacking circularisation or full gene annotation, yet they continue to be applied for numerous researchers without further verification. Therefore, such inappropriate use can lead to systematic bias in mitogenome-based phylogenetic reconstruction and functional interpretation, which is a matter of concern.

*Parastrongyloides trichosuri* mitogenome (LC050209) was originally deposited in the NCBI GenBank in 2016

TABLE I

Gene content, sequence length, and initiation/stop codons of *Parastrongyloides trichosuri* mitogenomes, incomplete mitogenome (Pt-i, NC_028620) and complete mitogenome (Pt-c, BK075097)

| Gene/region | | Positions and size (bp) | | Initiation and termination codons | | Anticodon |
|---|---|---|---|---|---|---|
| | | Pt-i[a] | Pt-c | Pt-i | Pt-c | Pt-c |
| 1 | *cox1* | 5,288-6,827 (1,540) | 1-1,560 (1,560) | ATG/TAG | ATT/TAA | |
| 2 | tRNA-Ala (A) | 6,828-6,882 (55) | 1,541-1,595 (55) | | | TGC |
| 3 | tRNA-Gln (Q) | 6,896-6,949 (54) | 1,609-1,662 (54) | | | TTG |
| 4 | *nad5* | 6,950-8,531 (1,582) | 1,690-3,285 (1,596) | TTG/TTT | ATT/TAA | |
| 5 | tRNA-Asn (N) | 8,532-8,590 (59) | 3,244-3,303 (60) | | | GTT |
| 6 | tRNA-Met (M) | 8,606-8,667 (62) | 3,319-3,380 (62) | | | CAT |
| 7 | tRNA-Val (V) | 8,668-8,722 (55) | 3,380-3,436 (57) | | | TAC |
| 8 | *nad6* | 8,723-9,160 (438) | 3,415-3,873 (459) | ATT/TAA | ATG/TAA | |
| 9 | tRNA-Pro (P) | 9,161-9,214 (54) | 3,874-3,927 (54) | | | TGG |
| 10 | tRNA-Tyr (Y) | 9,215-9,271 (57) | 3,928-3,984 (57) | | | GTA |
| 11 | *nad1* | 9,275-10,746 (870) | 3,982-4,857 (786) | ATT/TAG | TTG/TAG | |
| 12 | *atp6* | 10,147-10,746 (600) | 4,860-5,459 (600) | ATT/TAA | ATT/TAA | |
| 13 | tRNA-Lys (K) | 10,747-10,809 (63) | 5,460-5,522 (63) | | | TTT |
| 14 | tRNA-Arg (R) | 10,821-10,875 (55) | 5,534-5,588 (55) | | | GCG |
| 15 | *nad4L* | 10,876-11,107 (232) | 5,562-5,850 (289) | TTG/TTT | ATT/GGT | |
| 16 | tRNA-Trp (W) | 11,108-11,164 (57) | 5,821-5,877 (57) | | | TCA |
| 17 | tRNA-Phe (F) | 11,168-11,223 (56) | 5,881-5,936 (56) | | | GAA |
| 18 | tRNA-Gly (G) | 11,225-11,278 (54) | 5,938-5,991 (54) | | | TCC |
| 19 | cox2 | 11,225-11,974 (696) | 5,962-6,646 (685) | TTG/TAG | TTG/TAT | |
| 20 | tRNA-His (H) | 11,977-12,031 (55) | 6,690-6,744 (55) | | | GTG |
| 21 | *rrnL* | 12,032-12,985 (954) | 6,745-7,698 (954) | | | TCA |
| 22 | *nad3* | n.a. | 7,699-8,009 (311) | | TTG/TA[b] | |
| 23 | tRNA-Leu (L2) | 29-83 (55) | 8,549-8,631 (83) | | | TAA |
| 24 | tRNA-Ser (S1) | 84-137 (54) | 8,604-8,657 (54) | | | TTC |
| 25 | *nad2* | 138-978 (841) | 8,676-9,503 (828) | TTG/TTT | ATG/TAG | |
| 26 | tRNA-Ile (I) | 979-1,040 (62) | 9,499-9,560 (62) | | | GAT |
| 27 | tRNA-Cys (C) | 1,041-1,097 (57) | 9,561-9,617 (57) | | | GCA |
| 28 | tRNA-Asp (D) | 1,107-1,163 (57) | 9,627-9,683 (57) | | | GTC |
| 29 | *cytb* | 1,164-2,264 (1,101) | 9,657-10,784 (1,128) | TTG/TAA | TTG/TAA | |
| 30 | tRNA-Leu (L1) | 2,327-2,381 (58) | 10,847-10,901 (55) | | | AGA |
| 31 | *cox3* | 2,382-3,147 (766) | 10,902-11,669 (768) | TTG/ATT | TTG/TGA | |
| 32 | tRNA-Glu (E) | 3,148-3,202 (55) | 11,668-11,722 (55) | | | TTC |
| 33 | *rrnS* | 3,203-3,886 (684) | 11,722-12,408 (687) | | | |
| 34 | tRNA-Ser (S2) | 3,885-3,936 (52) | 12,407-12,458 (52) | | | AGA |
| 35 | tRNA-Thr (T) | 3,944-4,001 (58) | 12,466-12,523 (58) | | | TGT |
| 36 | *nad4* | 4,002-5,264 (1,263) | 12,545-13,786 (1,242) | TTG/TAA | ATG/TAA | |
| - | Non-coding region | 12,985-28 (743) | 8,010-8,548 (539) | | | |
| - | Repeat unit | n.a. | 8,269-8,343 (75) | | | |

a: the partial mitogenome (Pt-i) starts at 5,288 bp while the complete mitogenome (Pt-c) begins at 1 bp; b: non-canonical codons; n.a.: not available.

TABLE II

Pairwise distances rate (%) estimation of both individual and concatenated mPCGs and MRGs between *Parastrongyloides trichosuri* and members of the family Strongyloididae

| Species/strains[a] | Protein-coding genes (mPCGs) | | | | | | | | | | | | | Mitoribosomal genes (MRGs) | | |
|---|---|---|---|---|---|---|---|---|---|---|---|---|---|---|---|---|
| | atp6 | cox1 | cox2 | cox3 | cytb | nad1 | nad2 | nad3 | nad4L | nad4 | nad5 | nad6 | Con-mPCGs | rrnL | rrnS | Con-MRGs |
| *Strongyloides ratti* | 34.35 | 22.98 | **19.93** | 27.36 | **25.55** | 27.36 | **34.25** | **27.18** | **25.63** | 32.69 | **24.87** | **33.31** | **27.31** | 22.75 | **20.65** | **21.85** |
| *Strongyloides stercoralis* | **30.03** | 20.66 | 24.16 | 26.23 | 27.80 | **26.09** | 35.83 | 29.38 | 35.39 | **31.16** | 28.97 | 36.10 | 27.99 | **21.77** | 26.05 | 23.51 |
| *Strongyloides cebus* | 44.45 | **18.96** | 24.67 | 29.73 | 28.03 | 28.41 | 46.95 | 33.64 | 42.32 | 32.27 | 30.46 | 41.00 | 30.56 | 24.40 | 30.18 | 26.71 |
| *Strongyloides vituli* | 52.38 | 19.03 | 25.59 | 28.70 | 30.31 | 27.67 | 38.06 | 32.21 | 41.11 | 31.25 | 32.71 | 48.27 | 30.96 | 35.19 | 34.60 | 34.93 |
| *Strongyloides fuelleborni fuelleborni* (S) | 47.18 | 20.77 | 26.81 | **24.23** | 31.25 | 31.93 | 41.12 | 37.04 | 40.00 | 32.03 | 34.57 | 52.95 | 32.10 | 35.55 | 36.91 | 36.08 |
| *Strongyloides venezuelensis* | 46.93 | 22.86 | 33.36 | 28.78 | 30.11 | 27.62 | 41.71 | 34.08 | 35.73 | 34.47 | 30.93 | 48.69 | 32.18 | 30.30 | 35.58 | 32.39 |
| *Strongyloides fuelleborni fuelleborni* (L) | 55.88 | 23.85 | 29.79 | 31.20 | 35.90 | 34.47 | 47.20 | 42.26 | 51.90 | 35.08 | 40.62 | 58.91 | 36.89 | 36.55 | 38.51 | 37.35 |
| *Strongyloides papillosus* | 63.23 | 27.61 | 35.49 | 37.21 | 45.62 | 39.13 | 53.12 | 53.94 | 57.75 | 49.55 | 46.80 | 63.32 | 43.87 | 38.60 | 45.54 | 41.37 |

a: *Parastrongyloides trichosuri* (Pt-c, BK075097) served as the reference for pairwise distance rate calculations. The lowest value for each gene in the pairwise distance matrix was indicated in bold. mPCG, mitochondrial protein-coding gene; MRG: mitoribosomal gene; con-mPCGs: concatenation of 12 mPCGs; con-MRGs: concatenation of 2 MRGs.

and was designated as a RefSeq record (NC_028620) in 2023. Unfortunately, *P. trichosuri* mitogenome is incomplete, missing *nad3*, and has been misused as a complete mitogenome in several studies, including tRNA annotation,[24] and mPCGs-based phylogenomic inference.[4,25,26] Remarkably, Ko et al. noted that even the information providing regarding the source species and collection location was unreliable.[24] Taken together, these observations highlight the urgent need for stricter curation standards and critical assessment of public mitogenome data prior to reuse, underscoring the importance of the complete mitogenome and full annotation of *P. trichosuri* as a reliable reference for future mitogenomic studies.

In mitogenome analysis pipeline, while sample and sequencing quality are undoubtedly important, the choice of a specialised assembler and annotator is even more crucial. Among several assemblers, GetOrganelle toolkit[6] was employed as the best-performing assembler, as suggested by the benchmarking assembly studies.[7] To enhance assembly accuracy, a preprocessing step was applied to filter reads using parasite-specific mitochondrial sequences. Notably, in cestodes, the inclusion or omission of this step often determined the success or failure of the assembly.[22] In addition, this preprocessing step inherently helped to minimise host or pathogen contamination, thereby improving the overall assembly quality.

Within *Parastrongyloides* species, *P. trichosuri* exhibits distinct morphological characteristics, with spicules measuring 80-86 *μm* in length and having blunt, square-cut ends.[3] Smales et al.[3] also reported that *P. trichosuri* is most simlar to *S. stercoralis* (87.6%), followed by *S. ratti* (85.7%) and *S. procyonis* (85.7%) based on partial *cox1* (404 bp) sequence similarity. However, when the phylogeny of nine species was reconstructed using full-length *cox1* sequences, distinct clades were not clearly resolved, and *P. trichosuri* appeared to be more closely related to either *S. cebus* or *S. venezuelensis*, rather than to *S. stercoralis* within the Strongylidae, as shown in Fig. 3. This result highlights the limitations of using partial mitochondrial markers for species-level inference in nematodes. Comprehensive phylogenomic approaches employing full-length mitogenomes or con-mPCGs may provide more robust evolutionary resolution for *Parastrongyloides* and related taxa. Hunt et al. reconstructed the con-mPCGs-based phylogeny and found that *S. ratti* and *S. stercoralis*, as well as *S. papillosus* and *S. venezuelensis*, each formed distinct clades, with *P. trichosuri* being more closely related to the former clade.[2] Our mPCGs-based phylogenetic analysis showed a similar pattern.

Our phylomitogenomic analysis clearly distinguished two major clades, clade I (*P. trichosuri*, *S. cebus*, *S. stercoralis*, and *S. ratti*) and clade II (*S. venezuelensis*, *S. papillosus*, *S. vituli*, and *S. fuelleborni fuelleborni*). This pattern was consistent with the individual gene-based phylogenies derived from *atp6*, *nad2-4*, *nad4L*, and *nad6*, although the closest relative of *P. trichosuri* occasionally varied between *S. cebus* and *S. ratti*. The pattern of mitochondrial gene rearrangements was broadly congruent with the phylomitogenomic topology. Species within clade I displayed derived rearrangements,

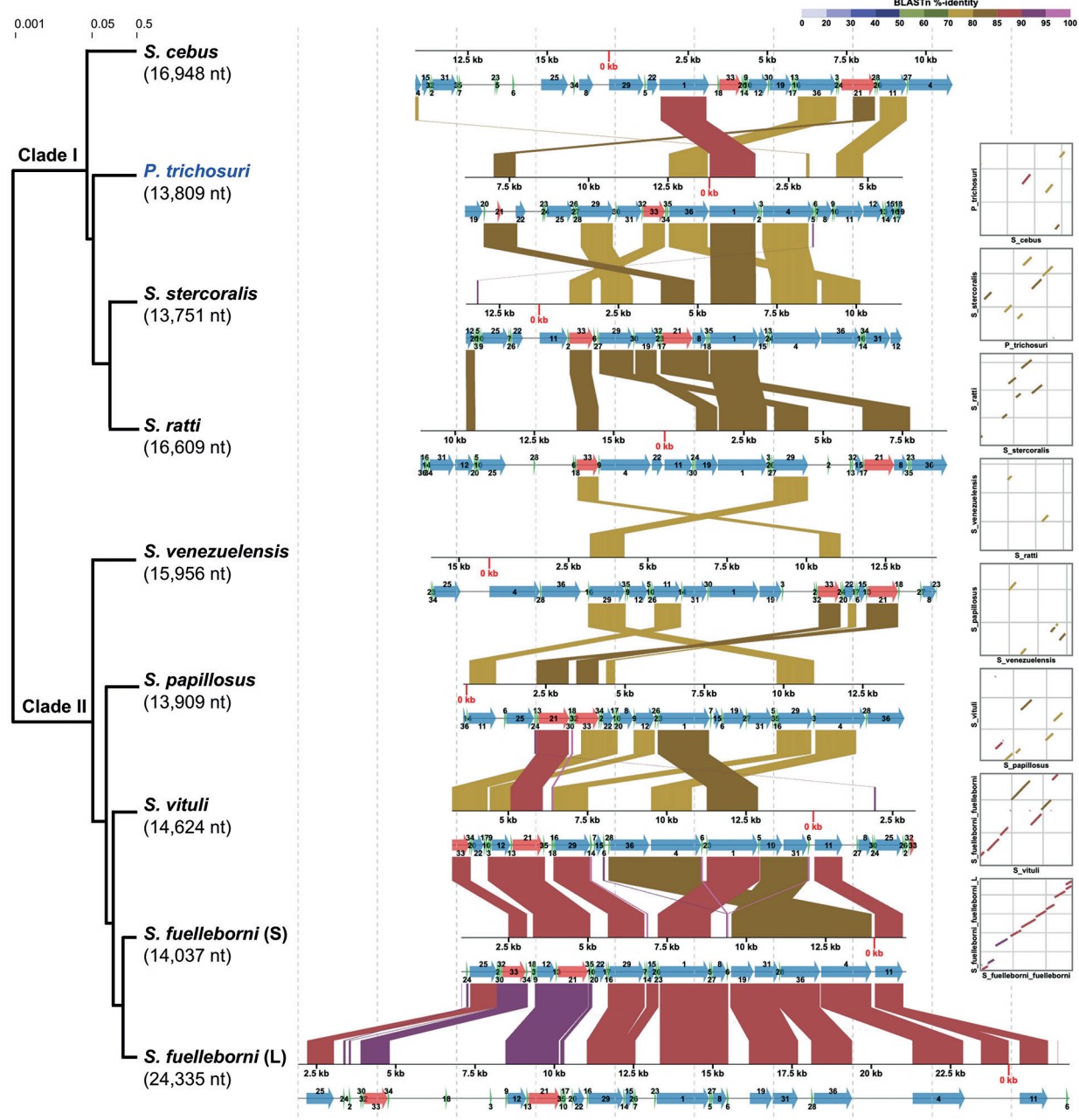

Fig. 2: micro-synteny and phylomitogenomic analyses of *Parastrongyloides trichosuri* and major *Strongyloides* species. Regions in adjacent assemblies are connected with coloured ribbons for similar regions in the alignment with Nucleotide %-identity (BLASTn %-identity) coded as colour from light blue to pink. All identified micro-syntenies were automatically aligned and positioned. Blue and red arrows on the assemblies depict the mPCGs and rRNA genes, respectively, while green arrowheads indicate tRNA genes. The rectangular phylogenetic tree is displayed with branch lengths scaled logarithmically. Each of the 36 genes is numbered, and the corresponding gene list is provided in Table I.

particularly involving *cox1* and *rrnL*, whereas those in clade II revealed rearrangements including *nad3*, *cytb*, *rrnS* and *rrnL*. Among these genes, *cytb* and *rrnS* exhibited extensive rearrangements spanning both clade I and clade II, whereas *cox1*, the most typical phylogenetic marker, was rearranged within each clade. Rather than *S. cebus*, *P. trichosuri* appears to have played a more pivotal role in the evolutionary history of the family Strongyloididae. These lineage-specific rearrangements offer key insights, suggesting that the evolution of mitochondrial gene order has been a major driver of diversification within this family.

*In conclusion* - We present an improved and complete mitogenome of *P. trichosuri*, featuring fully resolved gene annotations and a 539-bp NCR. This represents a notable improvement over the previously deposited RefSeq record (NC_028620), which lacks both the *nad3* gene and the NCR. Our findings underscore the importance of em-

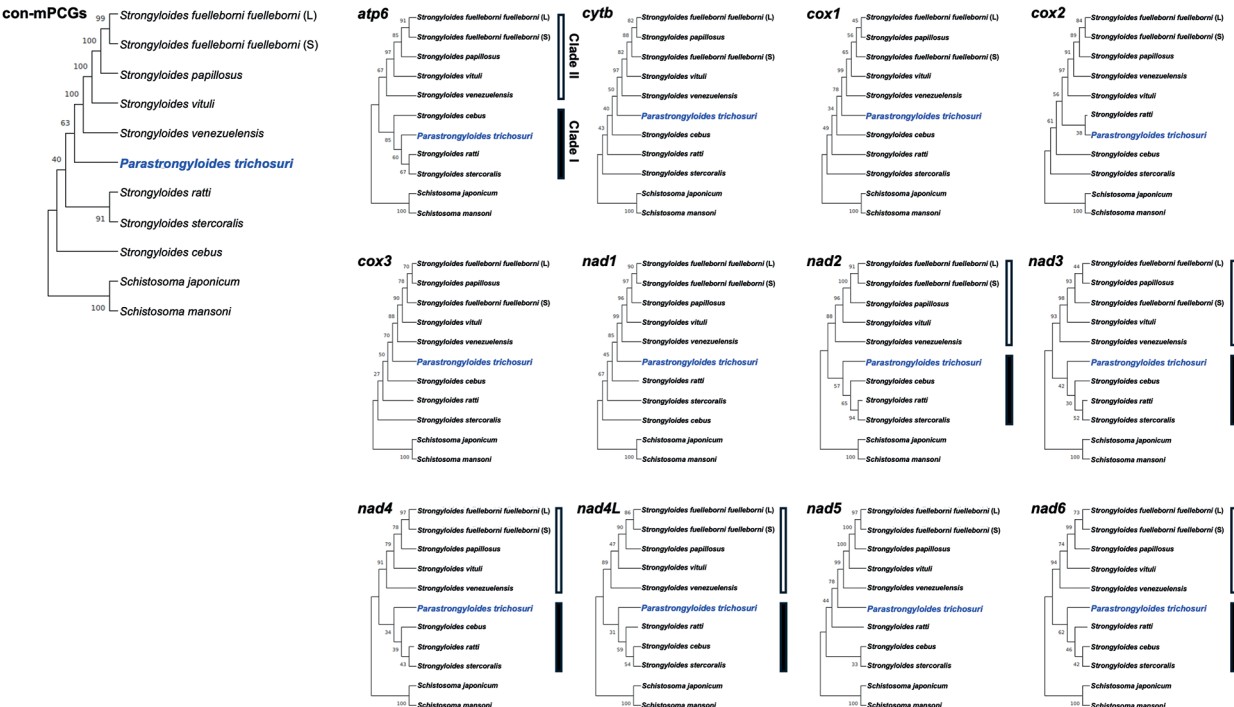

Fig. 3: phylogenetic trees showing the position of *Parastrongyloides trichosuri* (highlighted in bold blue). The trees are based on amino acid sequences derived from individual genes and from the concatenation of 12 mitochondrial protein-coding genes (con-mPCGs). *Schistosoma mansoni* and *S. japonicum* were used as outgroups. Nodal support values, estimated from 1,000 bootstrap pseudoreplicates, are indicated at each node. GenBank accession numbers are provided in Supplementary data (Table).

ploying a parasite-specific mitogenome assembly pipeline, which facilitates the reconstruction of high-quality mitogenomes from WGS datasets. The newly assembled *P. trichosuri* mitogenome provides a reliable and comprehensive resource for future comparative mitogenomic and phylomitogenomic investigations within the Strongyloididae and other related nematode lineages.

## ACKNOWLEDGEMENTS

To Mr Jun Won Song for his technical assistance with the installation and arrangement of the computing infrastructure and the anonymous reviewers whose comments and suggestions allowed us to improve the manuscript.

## AUTHORS' CONTRIBUTION

KC - conceptualisation, data curation, formal analysis, methodology, writing - original draft; MK - investigation, visualisation, writing - review & editing; JWP - investigation, writing - review & editing; Y-KC - conceptualisation, funding acquisition, writing - review & editing; WGY - conceptualisation, funding acquisition, methodology, supervision, visualisation, writing - original draft, writing - review & editing. All authors wrote, reviewed and approved the final version of the manuscript. The authors declare no conflicts of interest.

## DATA AVAILABILITY

The reassembled and reannotated complete mitogenome sequence data of *Parastrongyloides trichosuri* are available in the Third Party Assembly/Annotation Section of the DDBJ/ENA/GenBank databases under the accession number TPA: BK075097.

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

# OPEN PEER REVIEW

Memórias do IOC thanks the anonymous reviewers for their contribution to the peer review of this work.

## FIRST REVIEW ROUND

REVIEWERS' COMMENTS

### REVIEWER #1

This manuscript by Cho et al. titled "Mitochondrial genome refinement and comparative phylogenetics of Parastrongyloides trichosuri (KNP strain; Nematoda: Strongyloididae) from South Africa" is devoted to the refined mitochondrial genome (mitogenome) of Parastrongyloides trichosuri deposited in GenBank in 2016. The authors employ a rigorous and well-documented methodology for reassembly, reannotation, and validation. The manuscript is generally well-written, the data are robust, and the conclusions are supported by the results.

However, the authors have deposited the nearly complete mitogenome sequence in the Zenodo repository (DOI: 10.5281/zenodo.17351367). The Instructions for Authors for Memórias do Instituto Oswaldo Cruz clearly states the following regarding nucleotide sequences:

"Nucleotide sequences: Must be deposited in a public database (e.g., GenBank, EMBL, or DDBJ) before submission".

While Zenodo is an excellent general-purpose repository for supporting data, code, and entire manuscripts, the journal's policy explicitly mandates deposition in one of the three major, specialized, internationally recognized nucleotide sequence databases: GenBank, EMBL, or DDBJ. Zenodo is not listed as an acceptable alternative for primary nucleotide sequence data.

Therefore, the authors must deposit the final, annotated mitogenome sequence in GenBank, EMBL, or DDBJ and provide the corresponding accession number in the manuscript before it can be accepted for publication.

Minor points:

The authors refer to the mitogenome as "nearly complete". Please clarify what specific elements, if any, are missing or unresolved that prevent it from being considered fully complete.

Line 35: Replace "a existing" with "an existing"

Line 105: Specify the version of MAFFT used in the analysis.

### REVIEWER #2

This study reconstructs and refines the incomplete mitogenome of P. trichosuri, providing an accurate reference for comparative and phylogenetic studies within Strongyloididae. This study is well-designed, and the annotation of mPCGs is meaningful. The re-purification of bacterial DNA and effective utilization of existing sequencing data are also commendable. Therefore, the mitogenome presented in this manuscript (Zenodo DOI: 10.5281/zenodo.17351367) is expected to serve as a valuable reference sequence for future studies.

Minor issues to be updated:

1. Lines 113–114: Please clarify the rationale for choosing Schistosoma as the outgroup control.

2. Lines 173–176: Could you provide an example from nematodes rather than cestodes?

3. Line 212: Please check the italic formatting for Hunt et al. (2016).

AUTHORS' RESPONSE TO THE REVIEWERS

December 22, 2025

Point-to-point revisions

Manuscript ID: MIOC-2025-0294

Title: Mitochondrial genome refinement and comparative phylogenetics of Parastrongyloides trichosuri (KNP strain; Nematoda: Strongyloididae) from South Africa

We sincerely appreciate the reviewers' constructive feedback, which has substantially improved the quality of our manuscript. All comments have been carefully considered, and the corresponding revisions have been incorporated. Detailed responses to each point are provided below, and we hope that these revisions satisfactorily address all concerns.

Response to Reviewer 1:

- Comment 1: Therefore, the authors must deposit the final, annotated mitogenome sequence in GenBank, EMBL, or DDBJ and provide the corresponding accession number in the manuscript before it can be accepted for publication.

=> (response) Thank you for the insightful comment. Regarding the GenBank accession number for the complete mitochondrial genome of Parastrongyloides trichosuri, the sequence was originally submitted to NCBI GenBank on September 24, 2025, as a Third-Party Annotation (TPA) submission (Submission ID: 3006603). However, the assignment of an accession number was delayed due to the temporary U.S. federal government shutdown. We have now received the accession number (BK075097) and the previous Zenodo record has been replaced with the newly assigned accession number.

- Comment 2: The authors refer to the mitogenome as "nearly complete". Please clarify what specific elements, if any, are missing or unresolved that prevent it from being considered fully complete.
=> We appreciate the reviewer's valuable suggestion. There were no missing or unresolved elements in our assembly. However, we initially used cautious word because we did not perform additional validation procedures such as physical mapping or PCR-walking sequencing. Nevertheless, most studies analysing mitochondrial genomes using next-generation sequencing describe their sequences as "complete mitogenomes."[1,2] Accordingly, we have removed all expressions implying incompleteness.
[1] Ding C, Shen J, Ning S, Yang M 2025. The first complete mitochondrial genome of Dermestes vorax (Coleoptera: Bostrichiformia: Dermestidae) from China and its phylogenetic analyses. Mitochondrial DNA Part B 10: 283–287.
[2] Han XP, Song L, Wang Q, Chen XJ 2025. The complete mitochondrial genome and phylogenetic analysis of Syncrossus berdmorei (Blyth, 1860) (Cypriniformes: Botiidae). Mitochondrial DNA Part B 10: 376–381.

- Comment 3: Line 35: Replace "a existing" with "an existing"
=> (response) We have revised the manuscript in accordance with the reviewer's suggestion (Line 36).

- Comment 4: Line 105: Specify the version of MAFFT used in the analysis.
=> (response) We have revised the manuscript as suggested by the reviewer (Line 105).

Response to Reviewer 2:
- Comment 1: Lines 113–114: Please clarify the rationale for choosing Schistosoma as the outgroup control.
=> (response) We appreciate the reviewer's insightful comment. We selected Schistosoma spp. as the outgroup because they represent a taxonomically distant lineage (Platyhelminthes: Trematoda) relative to Parastrongyloides trichosuri (Nematoda), as suggested by Viney [3], who has also used non-nematode taxa as the outgroup in his/her analyses. This level of evolutionary distance is sufficient to root the tree without introducing excessive long-branch attraction. In addition, complete mitochondrial genome sequences of Schistosoma species are well-curated, stable, and frequently used as reliable outgroups in helminth phylogenetic studies, providing a consistent framework for comparative analyses. We have revised the manuscript in lines 114–116.
[3] Viney M 2017. How Can We Understand the Genomic Basis of Nematode Parasitism? Trends Parasitol. 33: 444–452.

- Comment 2: Lines 173–176: Could you provide an example from nematodes rather than cestodes?
=> (response) A nematode example has already been described immediately after this section (lines 178–183), and the relevant details are included in the Supplementary data (Table I).

- Comment 3: Line 212: Please check the italic formatting for Hunt et al. (2016).
=> (response) We have revised the manuscript as suggested by the reviewer (Line 213).

## SECOND REVIEW ROUND

REVIEWERS' COMMENTS

### REVIEWER #1

No comments.

### REVIEWER #2

No comments.

