## [Reviewer Report · FIRST REVIEW ROUND - REVIEWERS COMMENTS]

## REVIEWER #1

This manuscript by Cho et al. titled “Mitochondrial genome refinement and comparative phylogenetics of *Parastrongyloides trichosuri* (KNP strain; Nematoda: Strongyloididae) from South Africa” is devoted to the refined mitochondrial genome (mitogenome) of *Parastrongyloides trichosuri* deposited in GenBank in 2016. The authors employ a rigorous and well-documented methodology for reassembly, reannotation, and validation.

The manuscript is generally well-written, the data are robust, and the conclusions are supported by the results.

However, the authors have deposited the nearly complete mitogenome sequence in the Zenodo repository (DOI: 10.5281/zenodo.17351367).

The Instructions for Authors for *Memórias do Instituto Oswaldo Cruz* clearly states the following regarding nucleotide sequences:

“Nucleotide sequences: Must be deposited in a public database (e.g., GenBank, EMBL, or DDBJ) before submission”.

While Zenodo is an excellent general-purpose repository for supporting data, code, and entire manuscripts, the journal’s policy explicitly mandates deposition in one of the three major, specialized, internationally recognized nucleotide sequence databases: GenBank, EMBL, or DDBJ.

Zenodo is not listed as an acceptable alternative for primary nucleotide sequence data.

Therefore, the authors must deposit the final, annotated mitogenome sequence in GenBank, EMBL, or DDBJ and provide the corresponding accession number in the manuscript before it can be accepted for publication.

Minor points:

The authors refer to the mitogenome as “nearly complete”. Please clarify what specific elements, if any, are missing or unresolved that prevent it from being considered fully complete.

Line 35: Replace “a existing” with “an existing”

Line 105: Specify the version of MAFFT used in the analysis.

## REVIEWER #2

This study reconstructs and refines the incomplete mitogenome of *P. trichosuri*, providing an accurate reference for comparative and phylogenetic studies within Strongyloididae.

This study is well-designed, and the annotation of mPCGs is meaningful.

The re-purification of bacterial DNA and effective utilization of existing sequencing data are also commendable.

Therefore, the mitogenome presented in this manuscript (Zenodo DOI: 10.5281/zenodo.17351367) is expected to serve as a valuable reference sequence for future studies.

Minor issues to be updated:

1. Lines 113–114: Please clarify the rationale for choosing *Schistosoma* as the outgroup control.

2. Lines 173–176: Could you provide an example from nematodes rather than cestodes?

3. Line 212: Please check the italic formatting for Hunt et al. (2016).

## AUTHORS’ RESPONSE TO THE REVIEWERS

December 22, 2025

Point-to-point revisions

Manuscript ID: MIOC-2025-0294

Title: Mitochondrial genome refinement and comparative phylogenetics of *Parastrongyloides trichosuri* (KNP strain; Nematoda: Strongyloididae) from South Africa

We sincerely appreciate the reviewers’ constructive feedback, which has substantially improved the quality of our manuscript.

All comments have been carefully considered, and the corresponding revisions have been incorporated.

Detailed responses to each point are provided below, and we hope that these revisions satisfactorily address all concerns.

Response to Reviewer 1:

- Comment 1: Therefore, the authors must deposit the final, annotated mitogenome sequence in GenBank, EMBL, or DDBJ and provide the corresponding accession number in the manuscript before it can be accepted for publication.

=> (response) Thank you for the insightful comment. Regarding the GenBank accession number for the complete mitochondrial genome of *Parastrongyloides trichosuri*, the sequence was originally submitted to NCBI GenBank on September 24, 2025, as a Third-Party Annotation (TPA) submission (Submission ID: 3006603). However, the assignment of an accession number was delayed due to the temporary U.S. federal government shutdown. We have now received the accession number (BK075097) and the previous Zenodo record has been replaced with the newly assigned accession number.

- Comment 2: The authors refer to the mitogenome as “nearly complete”. Please clarify what specific elements, if any, are missing or unresolved that prevent it from being considered fully complete. => We appreciate the reviewer’s valuable suggestion.

There were no missing or unresolved elements in our assembly. However, we initially used cautious word because we did not perform additional validation procedures such as physical mapping or PCR-walking sequencing. Nevertheless, most studies analysing mitochondrial genomes using next-generation sequencing describe their sequences as “complete mitogenomes.”[1,2] Accordingly, we have removed all expressions implying incompleteness.

[1] Ding C, Shen J, Ning S, Yang M 2025. The first complete mitochondrial genome of *Dermestes vorax* (Coleoptera: Bostrichiformia: Dermestidae) from China and its phylogenetic analyses. Mitochondrial DNA Part B 10: 283–287.

[2] Han XP, Song L, Wang Q, Chen XJ 2025. The complete mitochondrial genome and phylogenetic analysis of *Syncrossus berdmorei* (Blyth, 1860) (Cypriniformes: Botiidae). Mitochondrial DNA Part B 10: 376–381.

- Comment 3: Line 35: Replace “a existing” with “an existing”

=> (response) We have revised the manuscript in accordance with the reviewer’s suggestion (Line 36).

- Comment 4: Line 105: Specify the version of MAFFT used in the analysis.

=> (response) We have revised the manuscript as suggested by the reviewer (Line 105).

Response to Reviewer 2:

- Comment 1: Lines 113–114: Please clarify the rationale for choosing *Schistosoma* as the outgroup control.

=> (response) We appreciate the reviewer’s insightful comment. We selected *Schistosoma* spp. as the outgroup because they represent a taxonomically distant lineage (Platyhelminthes: Trematoda) relative to *Parastrongyloides trichosuri* (Nematoda), as suggested by Viney [3], who has also used non-nematode taxa as the outgroup in his/her analyses. This level of evolutionary distance is sufficient to root the tree without introducing excessive long-branch attraction. In addition, complete mitochondrial genome sequences of *Schistosoma* species are well-curated, stable, and frequently used as reliable outgroups in helminth phylogenetic studies, providing a consistent framework for comparative analyses. We have revised the manuscript in lines 114–116.

[3] Viney M 2017. How Can We Understand the Genomic Basis of Nematode Parasitism? Trends Parasitol. 33: 444–452.

- Comment 2: Lines 173–176: Could you provide an example from nematodes rather than cestodes?

=> (response) A nematode example has already been described immediately after this section (lines 178–183), and the relevant details are included in the Supplementary data (Table I).

- Comment 3: Line 212: Please check the italic formatting for Hunt et al. (2016).

=> (response) We have revised the manuscript as suggested by the reviewer (Line 213).